# Effect of Hypothermia Therapy on Children with Traumatic Brain Injury: A Meta-Analysis of Randomized Controlled Trials

**DOI:** 10.3390/brainsci12081009

**Published:** 2022-07-30

**Authors:** Qiujing Du, Yuwei Liu, Xinrong Chen, Ka Li

**Affiliations:** West China School of Nursing/West China Hospital, Sichuan University, Chengdu 610041, China; duqiujing001@163.com (Q.D.); liuyuwei@wchscu.cn (Y.L.); chenxinrong0504@126.com (X.C.)

**Keywords:** traumatic brain injury, hypothermia, children, meta-analysis

## Abstract

Hypothermia therapy is a promising therapeutic strategy for traumatic brain injury (TBI); however, some trials have shown that hypothermia therapy has a negative effect on patients with TBI. The treatment of hypothermia in children with TBI remains controversial. We conducted a search of six online databases to validate the literature on comparing hypothermia with normal therapy for children with TBI. Eight randomized controlled trials (514 patients) were included. The meta-analysis indicated that hypothermia therapy may increase the Glasgow Outcome Scale (GOS) scores. However, in terms of improving the rate of complications, intracranial pressure (ICP), mortality, cerebral perfusion pressure (CPP), and length of stay both in hospital as well as pediatric ICU, the difference was not statistically significant. Hypothermia therapy may have clinical advantages in improving the GOS scores in children with TBI compared with normothermic therapy, but hypothermia therapy may have no benefit in improving the incidence of complications, ICP, mortality, CPP, and length of stay both in pediatric ICU as well as hospital. The decision to implement hypothermia therapy for children with TBI depends on the advantages and disadvantages from many aspects and these must be considered comprehensively.

## 1. Introduction

TBI is the organic damage to brain tissue because of external force acting directly or indirectly on the head [1]. There is a major issue in global public health, pediatric TBI, which affects millions of children every year [2]. Approximately 3 million children worldwide are affected by TBI each year [3]. In the United States, pediatric TBI is one of main reasons for death as well as disablement among children [4]. On the basis of statistics, there are 1 million children injured by TBI in the United States every year, among which 85–90% are mild traumatic brain injury [5,6] which has serious adverse effects on children’s short-term and long-term functions [7]. Primary TBI brain injury includes skull fracture, brain contusion, laceration, hematocele, and diffusing axonal damage, leading to most nonreversible brain injuries; secondary TBI may be caused by endocranial or extracranial influences. Intracranial consequences include lumpy lesions, focal/diffuse brain swelling, intracranial hypertension, seizures, vasospasm, or infection, while extracranial consequences include hypotension, hypoxia, hypercapnia/hypocapnia, hyperglycemia/hypoglycemia, anemia, fever, electrolyte abnormalities, coagulation disorders, and infection [8]. There are no effective medications for TBI, and few measures other than prevention can mitigate the primary injury, whereas the “rolling” pathology of delaying secondary injury allows for intervention within a limited time frame [9]. Thus, TBI may include prevention strategies as well as therapy for secondary brain damage.

It has been reported that hypothermia can be applied to remedy TBI. In the 1990s, a TBI animal model proved the protective effect of hypothermia on brain tissue [10]. Similar effects of hypothermia on severe TBI have been demonstrated in small clinical trials, but not in large trials [8]. The specific effect of hypothermia is to restrict secondary brain damage through decreasing intracranial pressure (ICP) as well as brain metastasis demand, reducing the destruction of the blood–brain barrier, restraining inflammation cytokines as well as decreasing free radicals associated with reperfusing damage [11,12]. However, the results of a few randomized controlled trials showed that hypothermia treatment of patients with intracranial hypertension after TBI did not improve their outcome [13,14,15].

In recent years, more and more studies have explored the influence of TBI on children [16]. Thermoregulation is a complex process that is essential for homeostasis and survival [17]. It is coordinated by the hypothalamus’s thermoregulatory center. The autonomic thermoregulatory response of children is lower than that of normal adults due to the hypoplasia of thermoregulatory centers and complex changes in synaptic development, metabolism, and blood flow [18,19]. Therefore, the effect of hypothermia on TBI children is still controversial.

We found that a Bayesian analysis of hypothermia was used in children with severe TBI, but the analysis only discussed outcome indicators of mortality. The results showed that treatment with hypothermia was one-third more likely to reduce the relative risk of death by more than 20% compared with treatment with normal temperature. The advantage of this meta-analysis lies in the exploration of multiple clinical outcome indicators of TBI in children treated with hypothermia and the inclusion of more high-quality randomized controlled trials to provide theoretical support for clinical decision making. 

## 2. Methods

The study was authorized by the Ethics Committee on Biomedical Research, West China Hospital of Sichuan University (2021-0604). System evaluation and meta-analysis were conducted according to the preferred report items of system evaluation and meta-analysis guidance (checklist in non_published_material). The study protocol was registered at the PROSPERO (registration number: CRD42022324579).

### 2.1. Information Sources and Search Strategies

We performed a systematic document search in PubMed, Ovid Embase, Web of Science, the Cochrane Library, and Wanfang Database, as well as China National Knowledge Infrastructure to verify related research issued in Chinese as well as English from the inception of every database through April 2022. The search used a combination of the following terms: hypothermia, low temperature, moderate hypothermia, baby, infant, kid, pediatric, children, traumatic brain injury, head injury, and brain injuries. Searching was performed via title/abstract, key words, as well as medical subject headings (MeSH) terminologies. Moreover, the underlying related research in reference lists was tested. For example, the electronic searching tactics in the PubMed database were as follows: ((((moderate hypothermia) OR (low temperature)) OR (hypothermia[Title/Abstract])) AND (((((baby[Title/Abstract]) OR (infant)) OR (kid)) OR (pediatric)) OR (children))) AND (((traumatic brain injury[Title/Abstract]) OR (head injury)) OR (brain injuries)). Filters: Humans; Chinese; English.

### 2.2. Inclusion Standard as Well as Exclusion Standard

Research that met the standards below were included: (1) research design: randomized controlled trial (RCT); (2) study population: children having traumatic brain injury (less than 18 years old and hospitalized 8 h after injury); (3) intervention: hypothermia therapy; (4) comparison intervention: normothermic therapy; and (5) outcome measure: at least 1 of the following was evaluated: ① primary outcome: mortality, occurrence rate of complications (infection, arrhythmia, coagulation disorder, etc.); ② secondary outcome: intracranial pressure (ICP), cerebral perfusion pressure (CPP), Glasgow Outcome Scale (GOS) scores, Pediatric Intensive Care Unit (PICU) length of stay, stay length in hospital. Exclusion standards were (1) there was no full text; (2) the language of publication was not English or Chinese; (3) data cannot be extracted; and (4) inconsistent result indexes. Two independent investigators (Q.D. and Y.L.) performed document filtrating and information extractions well as crosschecking. The disagreement was thrashed out through a discussion.

### 2.3. Data Extraction and Quality Evaluation

Data extraction from each study was conducted independently by two authors (QJ.D. and YW.L.). Any differences were settled through a discussion with the third author (K.L.). The following items were drawn from the research: (1) research features: first author, publication year, as well as state; (2) participators: number of patients (intervention/control), age, Glasgow Coma Score (GCS), and location of body temperature measurement; (3) intervention: the method of hypothermia therapy, target temperature; (4) control: target temperature; and (5) outcome measure: mortality, incidence of complications (infection, arrhythmia, coagulation disorder, etc.); ICP, CPP, GOS scores, stay length in PICU, stay length in hospital. Again, differences in data extraction were resolved by consensus. Endnote Version 8.0 software was used for literature retrieval, extraction, management, and citation.

All contained research was evaluated against the standard of the Cochrane Handbook of Systematic Evaluation of Interventions [20], with a detailed list to assess bias risking. The following items below were evaluated as a low, high, or uncertain bias risk: randomized generating, allocation hiding, blinding (participators, researchers, and result evaluators), incomplete result information, optional result report, as well as other biases [20]. Similarly, quality evaluation was carried out by two independent reviewers (QJ.D. YW.L). Uncertainties or differences were settled by negotiating, and the third researcher (K.L) conducted quality control throughout the process. 

### 2.4. Data Synthesis

Statistical analysis was performed by Review Manager 5.3 software (The Nordic Cochrane Centre, Copenhagen, Denmark). Continuous outcome data were expressed as the mean difference with 95% CI when all studies were of the same unit and magnitude; if not, the standard mean difference was used instead. For dichotomous result information, relative risking (RR) with 95% CI was applied for assessment. 

Statistical heterogeneity was evaluated with a Q test as well as I^2^ measurement, and non-homogeneity was thought evident if I^2^ was greater than 50% and the *p* value was less than 0.10. Researchers often accumulate data from a series of studies performed by independent researchers, but it is unlikely that all the studies were functionally equivalent, so we performed a random-effects meta-analysis for each outcome. If there was significant heterogeneity among studies, sensitivity studies were conducted to further explore the source of heterogeneity.

Meanwhile, a sensitivity study was conducted to exclude every paper in turn and explore the influence of each study on the total effect. Finally, forest plots were produced to evaluate the impact of result variates for all research and to depict the statistical outcomes of the meta-analysis. A *p*-value of <0.05 represented statistical significance. 

## 3. Results

### 3.1. Features of Included Researches

In the meta-analysis, 940 independent studies were originally verified in electronic databases, after 266 copies were removed. A total of 53 studies qualified for a further full text search, of which 45 studies did not satisfy the included standard, which left eight studies (with 514 patients) contained in the final study. The flow diagram with specific data is summarized in Figure 1 [21,22,23,24,25,26,27,28].

### 3.2. Study Features

The features of the contained items are shown in Table 1.

### 3.3. Study Designs

Of all the included studies, six were single center studies and two were multicenter studies. A total of seven studies were published in English and only one study was published in Chinese.

### 3.4. Participants

Overall, 514 patients were selected aged from 0 to 18 years old. The sample size ranged from 21 [24] to 225 [25]. A total of six studies measured body temperature in the rectum and two in the esophagus. The GCSs of patients in seven studies were less than 8 points, and the GCSs of patients in one study [28] were not reported.

### 3.5. Interventions and Controls

All the trials were conducted to compare hypothermia with normal temperature treatment alone. The temperature of hypothermia treatment is 32–35 °C, and the temperature of normal temperature treatment is 36–38.5 °C.

### 3.6. Outcome Measures

A total of seven studies reported mortality, four studies reported the incidence of complications, five studies reported ICP, three studies reported CPP, two studies reported GOS scores, three studies reported stay length of PICU, and three studies reported stay length of hospital.

### 3.7. Quality Evaluation 

The method quality for eight contained studies is shown in Figure 2. All studies reported specific methods of randomized order generation and allocation concealment; thus, corresponding domains were assessed as “low risk”. A total of two trials [20,23] reported specific methods of the blinding of participators and staff and the blinding of the result assessment; thus, corresponding domains were assessed as “low risk”. None of the patients described in the study dropped out or dropped out of this experiment. As a result, the entrance was assessed as “low risk.” Eight trials did not clearly report appropriate methods of selective reporting and other biases. Due to insufficient information, we believe that there was an uncertain risk of deviation between the two projects. Other sources of bias were not evaluated in the meta-analysis.

### 3.8. Synthesis Results of the Primary Outcome

For mortality (%), the random influences model was applied and no significant non-homogeneity was explored in the research (I^2^ = 0%, *p* = 0.43). The meta-analysis indicated that the hypothermia therapy part had a higher mortality than the normothermic therapy part (Figure 3a), but the difference was not statistically significant (*p* = 0.07). The incidence of complications was a random-influence model, and no significant heterogeneity was found in the study (I^2^ = 41%, *p* = 0.16). The meta-analysis indicated that the hypothermia therapy part had a lower occurrence rate of complications than the normothermic therapy part (Figure 3b); however, the distinction was not evident in the statistics (*p* = 0.77). The sensitivity study showed that the consequences of the two indications were reliable and did not rely on any single study. 

### 3.9. Synthesis Results of the Secondary Outcome

For intracranial pressure (ICP) (mmHg), a randomized influences model was applied, and significant non-homogeneity was explored in the research (I^2^ = 96%, *p* < 0.00001). Subsequently, a sensitivity study was conducted, and after excluding Li et al. [25], non-homogeneity in the research reduced from I^2^ = 96% to I^2^ = 0%, suggesting that the research was originally non-homogeneous. The meta-analysis indicated that the hypothermia therapy part had a lower ICP than the normothermic therapy part (Figure 4a), but the difference was not statistically significant (*p* = 0.37). Moreover, for the cerebral perfusion pressure (CPP) (mmHg), a randomized influences model was applied, and no significant non-homogeneity was explored in the research (I^2^ = 0%, *p* = 0.81). The meta-analysis showed that the CPP of hypothermia was higher than that of the normal treatment, but the difference was not statistically significant (*p* = 0.21) (Figure 4b). The sensitivity study indicated that the consequences of this indicator were reliable and did not rely on any single study. 

For the Glasgow Outcome Scale (GOS scores) (points), a randomized influences model was applied, and significant non-homogeneity was explored in the research (I^2^ = 73%, *p* = 0.05). The meta-analysis indicated that the hypothermia therapy part had higher GOS scores than the normothermic therapy part (Figure 5a), and the difference was statistically significant (*p* = 0.01). Moreover, for the stay length in PICU (days) as well as the stay length in hospital (days), a randomized influences model was applied to conduct a combined study without evident heterogeneity in the research (PICU: I^2^ = 15%; hospital length of stay: I^2^ = 0%). The meta-analysis showed that the length of hospital stay (days) and length of hospital stay (days) of PICU in patients treated with hypothermia were higher than those in normal treatment group (Figure 5b–c); however, the distinction was not evident in the statistics. (PICU: *p* = 0.48; stay length in hospital: *p* = 0.93). The sensitivity study indicated the consequences of the three indicators were reliable and did not rely on any single study.

## 4. Discussion

The purpose of the systematic review and meta-analysis was to explore the effects of hypothermia on children with TBI. In this review, we identified eight randomized controlled trials comparing hypothermia to normal temperature therapy. In general, compared with normothermic treatment, hypothermia may improve GOS scores. However, in terms of improving the rate of complications, intracranial pressure (ICP), mortality, cerebral perfusion pressure (CPP), and length of stay both in hospital as well as pediatric ICU, the difference was not statistically significant.

TBI is one of the major reasons of death as well as disability among children and is a main universal global public health issue [24]. Children with serious TBI (Glasgow Coma Score (GCS) < 9) usually have serious and persistent neurocognition and absence. Average IQ scores were18–26 points (1–2 standard deviations) lower than the control group [29]. The impacts of TBI on individuals, families, societies, and economies are far-reaching. In the United States, there are about 1.7 million TBI patients and about 50,000 deaths each year. Every year the economic burden of TBI in United States is calculated to be USD 76.5 billion [30].

Secondary brain injury, leading to damaged self-regulation, systemic hypotension, and brain ischemia as well as encephalic hypertension, is the main element affecting the prognosis of TBI patients [27]. The purpose of drug treatment is to minimize the secondary damage [31,32]. The cool therapy known as hyperthermia has the potential to treat the multiple pathological effects of central nervous system injuries [33]. Mild hypothermia could decrease secondary brain damage through decreasing brain ischemia, encephaledema, and as tissue damage via damping exciting aminoacids generation [34,35,36]. Studies have also shown that therapeutic hypothermia (TH) has advantageous influences on all sides on the secondary brain damage of animal models [37,38]. In humans, studies have shown that the usage of mild hypothermia in the treatment of brachychronic TBI can improve the prognosis of adult patients [25]. 

The results of one previously published article [23] showed that therapeutic hypothermia did not increase any complications, including infections, bleeding, or arrhythmias, compared to normothermia, and a slow rewarming was carried out after 72 h of hypothermia without compromising CPP significantly, which indicates that therapeutic hypothermia may be safe in the treatment of traumatic brain injury in children. This review also did not unexpectedly find benefits of hypothermia therapy in children with TBI, such as it may improve the GOS scores, which may be related to the mechanism described above. However, this review did not find that hypothermia improved the rate of complications, intracranial pressure (ICP), mortality, cerebral perfusion pressure (CPP), and length of stay both in hospital as well as pediatric ICU, which is consistent with the findings of Tasker, R.C., et al. [39], although they only discussed the outcome indicator of mortality. Indeed, high-quality trials have shown that therapeutic hypothermia has a neutral or even negative impact on long-term outcomes [40,41,42]. One possible explanation is that hypothermia itself is a risk factor for negative effects in trauma patients (e.g., mortality, higher CPP, etc.) [43]. Meanwhile, these outcomes may be influenced by hemodynamics in management and care [42]. In addition, all aspects of hypothermia treatment, including induction time, duration and depth, rewarming rate, different institutions, and protective effects of hypothermia treatment on TBI patients may change due to these factors [10]. Therefore, whether to implement hypothermia treatment for children with TBI depends on the advantages and disadvantages of various aspects and comprehensive consideration needs to be taken. 

RCTs are generally considered the gold standard for clinical studies. Randomized controlled trials are usually limited by results and time and often only ensure the statistical validity of primary outcome measures. Because of severe inclusive as well as exclusive standards, the sample size of RCTs is often small, leading to greater bias. However, the RCTs included in this study were of high quality, which increases the credibility of the results of this study. Due to the heterogeneity of the meta-analysis due to different implementation methods of each study (such as the variability of hypothermia treatment methods and target temperature, etc.), we performed a randomized influence meta-analysis for each single result. For studies with obvious inhomogeneity, sensitivity studies were conducted to further explore the source of the inhomogeneity. At the same time, a sensitivity analysis was conducted to exclude each paper in turn and to find out the effect of each study on the overall effect. We discovered that the consequences were reliable and did not rely on any single study. However, for the mortality, complication rate, ICP, CPP, stay length in PICU, as well as the stay length in hospital, a difference between two groups was not evident in the statistics. The reasons for this result could be because of the small sample size and inconsistency between hypothermia treatment (variability of hypothermia treatment and target temperature) and normal temperature treatment (variability of target temperature). 

The research inevitably has some limitations. Firstly, although the interventions in the study were uniform, they were not all the same. Therefore, there must be some clinical inhomogeneity, which has a certain impact on the results of the study. Secondly, this study only included original research published in Chinese and English. Language restrictions could produce selective bias, which impacts the dependability of combinatorial consequences. Thirdly, six of contained studies did not depict the implementation of blinding. There could be biases that would impact the ultimate consequences, showing that the study’s methodology has some restrictions. Fourth, among the eight randomized controlled trials included in this study, five trials were conducted on children with severe traumatic brain injury, and the remaining three trials did not report the types of traumatic brain injury in children. Due to the complex and changeable condition of patients with severe traumatic brain injury, this may have biased the research results, that is, it may have underestimated the effectiveness of hypothermia treatment. Up to now, whether to implement hypothermia therapy for children with TBI is still controversial. This meta-analysis provides a good reference for solving this problem. Therefore, this meta-analysis was necessary and has important clinical guiding significance. According to the present problems in the current research, increasingly rigorous RCTs, containing multicenter, placebo-controlled clinic trials are required to generated higher quality proof. A suitable random approach and sample size estimation were used. As for trial reporting, the investigator needs to follow a detailed list of the Comprehensive Standards for Trial Reports [44].

## 5. Conclusions

Existing evidence suggests that hypothermia is superior to normothermic therapy in improving GOS scores in children with TBI, but it may have no significant impact on improving the incidence of complications, ICP, mortality, CPP, length of hospital stay, and length of hospital stay in PICU. More well-designed and high-quality RCTs are needed to further evaluate the effectiveness of hypothermia therapy in children with TBI and to provide reasonable theoretical guidance for clinical practice. 

## Figures and Tables

**Figure 1 brainsci-12-01009-f001:**
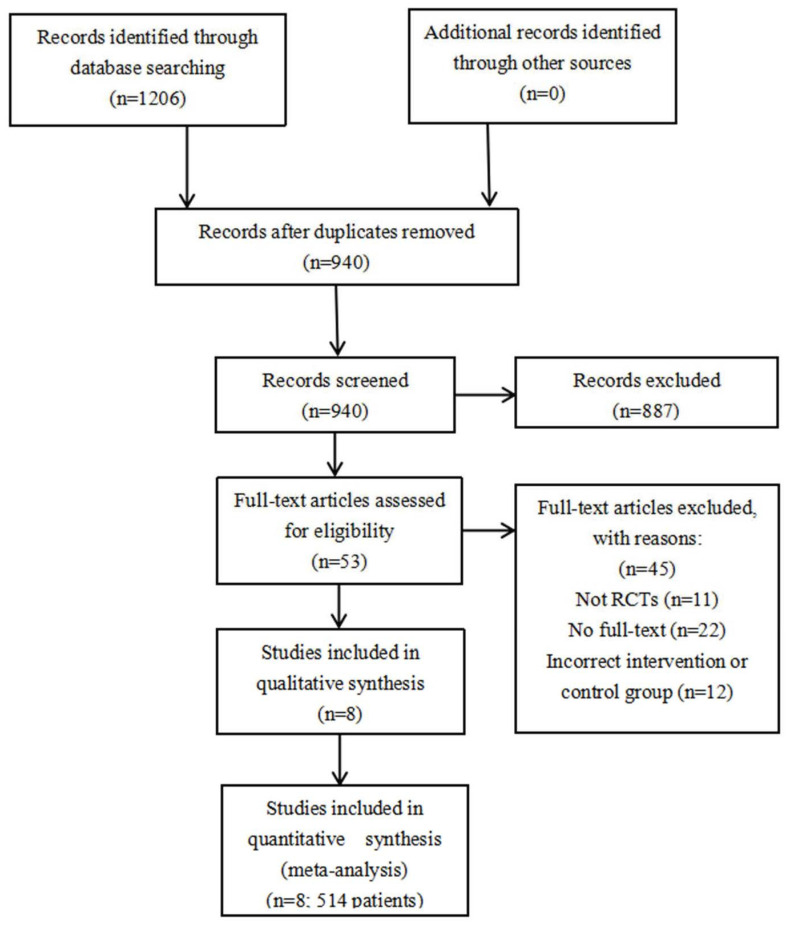
Flow chart of selection of included studies. RCT: randomized controlled trial.

**Figure 2 brainsci-12-01009-f002:**
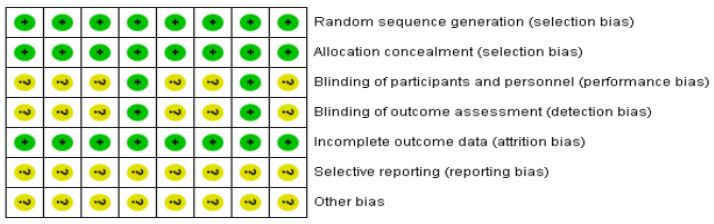
Risk-of-bias assessment based on review author’s judgment about the risk-of-bias item for each eligible study (*n* = 8). +, low risk of bias; ?, unclear risk of bias [21,22,23,24,25,26,27,28].

**Figure 3 brainsci-12-01009-f003:**
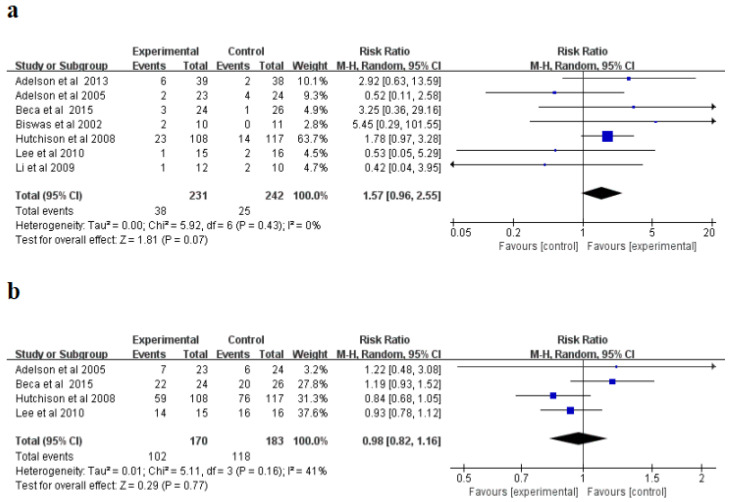
Forest plots of the meta-analysis of mortality (**a**), and incidence of complications (**b**) M-H, mantel-haensze [21,22,23,24,25,26,27].

**Figure 4 brainsci-12-01009-f004:**
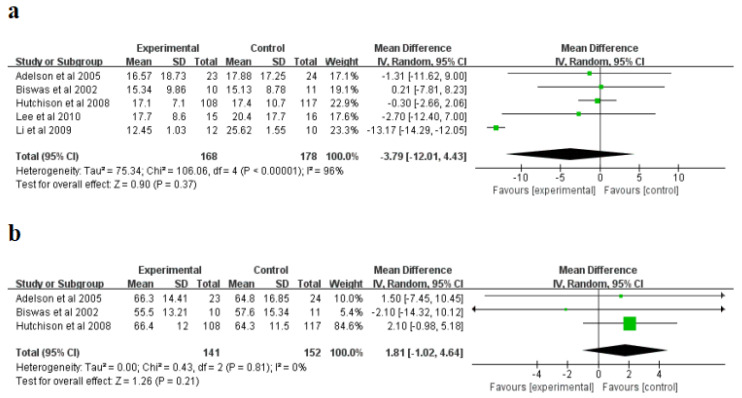
Forest plots of the meta-analysis of intracranial pressure (ICP) (**a**), and cerebral perfusion pressure (CPP) (**b**). IV, inverse variance [21,24,25,26,27].

**Figure 5 brainsci-12-01009-f005:**
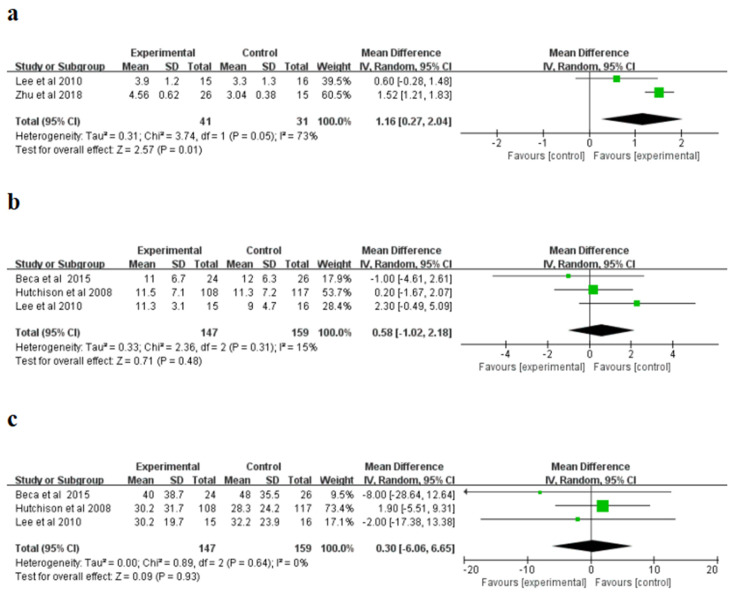
Forest plots of the Glasgow Outcome Scale (GOS) (**a**), Pediatric Intensive Care Unit (PICU) length of stay (**b**), and hospital length of stay (**c**). IV, inverse variance [23,25,26,28].

**Table 1 brainsci-12-01009-t001:** Main characteristics of studies included in the meta-analysis.

Study ID	Country	Sample Size (Intervention/Control)	Age	location of Body Temperature Measurement	Glasgow Coma Score (GCS)	Intervention (the Method of Hypothermia Therapy, Target Temperature)	Control (Target Temperature)	Outcome
Adelson et al., 2005 [21]	The United States	47 (23/24)	0–17 years old	Rectum	5–8	Use the cooling blanket for 48 h and reduce the temperature to 32–33 °C	Maintain body temperature at 36.5–37.5 °C	①②③④
Adelson et al., 2013 [22]	The United States	77 (39/38)	0–17 years old	Rectum	5–7	Use the cooling blanket for 48 h and reduce the temperature to 32–33 °C	Maintain body temperature at 36.5–37.5 °C	①
Beca et al., 2015 [23]	Australia, New Zealand and Canada.	50 (24/26)	1–16 years old	Esophagus	3–7	Use the cooling blanket for 72 h and reduce the temperature to 32–33 °C	Maintain body temperature at 36–37 °C	①⑥⑦
Biswas et al., 2002 [24]	The United States	21 (10/11)	0–18 years old	Rectum	3–7	Use the cooling blanket for 48 h and reduce the temperature to 32–34 °C	Maintain body temperature at 36.5–37.5 °C	①③④
Hutchison et al., 2008 [25]	Canada, England, and France	225 (108/117)	1–17 years old	Esophagus	3–6	Use the surface cooling techniques for 24 h and reduce the temperature to 32.5 ± 0.5 °C	Maintain body temperature at 37 ± 0.5 °C	①②③④⑥⑦
Lee et al., 2010 [26]	Taiwan, China	31 (15/16)	0–12 years old	Rectum	4–8	Use the cooling blanket and reduce the temperature to 32–35 °C	Maintain normal body temperature	①②③⑤⑥⑦
Li et al., 2009 [27]	China	22 (12/10)	6–108 months old	Rectum	<8	Use the cooling cap for 72 h and reduce the temperature to 34.5 ± 0.2 °C	Maintain body temperature at 37.5–38.5 °C	①③
Zhu et al., 2018 [28]	China	41 (26/15)	1–14 years old	Rectum	NR	Use the cooling blanket for 3–7 days and reduce the temperature to 33–35 °C	Maintain normal body temperature	⑤

①, mortality; ②, incidence of complications; ③, intracranial pressure (ICP); ④, cerebral perfusion pressure (CPP); ⑤, Glasgow Outcome Scale (GOS); ⑥, Pediatric Intensive Care Unit (PICU) length of stay; ⑦, hospital length of stay. NA: not reported.

## Data Availability

No new data were created or analyzed in this study. Data sharing is not applicable to this article.

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
