# Peer review of "Effect of Hypothermia Therapy on Children with Traumatic Brain Injury: A Meta-Analysis of Randomized Controlled Trials"

_brainsci, 2022, doi:10.3390/brainsci12081009_

Round 1
Reviewer 1 Report
Extensive English editing is required prior to publications. Examples of erroneous English is cerebrum (reference to only the telencephalon), which is restrictive and does not include whole brain TBI injury, reference to metastasis, rather than metabolism.
Are the Authors able to separate out adverse events such as coagulopathies etc?
Can the Authors separate analysis based on types of TBI?
In section 3.8. mortality should not be reported as a percentage, should be absolute number
Please add titles to Forrest plots to improve readability.
It is not surprising that hypothermia results in higher GOS and length of PICU stay given most patients are intubated for up to 72h. This is an accepted outcome and is a result of additional time required to administer the therapy as opposed to a negative therapeutic outcome.
It is not wise for the authors to highlight in the Discussion that hypothermia translates to higher mortality and CPP, given the value is not significant. This should be avoided or significantly toned down.
Can the Authors evaluate additional neurological outcome improvement as a secondary outcome. Currently, only acute outcomes are reported, which does not necessarily investigate the efficacy of hypothermia in pediatric TBI
Reviewer 2 Report
The manuscript presented by Du et al., entitled "Effect of hypothermia therapy on children with traumatic brain injury: a meta-analysis of randomized controlled trials" is interesting. However there are several major points to improve the quality of this:
- The author should extend the introduction, for example they mention only statistic report from USA, they should also report Europe and Asia.
- The author should improve the quality of the results, It seems that they are screenshot and the resolution of table is not clear. The author should insert specific table.
- The author should also extend the material and method.
- The author should discuss the difference of the results reported in previous published article:
Beca, J., McSharry, B., Erickson, S., Yung, M., Schibler, A., Slater, A., Wilkins, B., Singhal, A., Williams, G., Sherring, C., Butt, W., & Pediatric Study Group of the Australia and New Zealand Intensive Care Society Clinical Trials Group (2015). Hypothermia for Traumatic Brain Injury in Children-A Phase II Randomized Controlled Trial. Critical care medicine, 43(7), 1458–1466. https://doi.org/10.1097/CCM.0000000000000947
Reviewer 3 Report
This meta-analysis aims at assessing the interest of hypothermia treatment in pediatric TBI patients. We have minor to moderate comments: - the quality of the English should be improved overall. - in the introduction : the authors repeat themselves in the end of the section. - in the methods: information sources and search strategies: it would be helpful to have the entire search terms in an appendix - in the results: a) the outcome in table 1 should be mentioned explicitly instead of numbers since it might be confusing for the reader; b) since this analysis includes 514 pediatric cases between 0-18 years, would it be possible to extract individual data to better understand the impact of treatment on more precise age ranges?
Round 2
Reviewer 2 Report
The authors satisfied all my concerns.